# Deep Biological Pathway Informed Pathology-Genomic Multimodal Survival Prediction

## Abstract

The integration of multi-modal data, such as pathological images and genomic data, is essential for understanding cancer heterogeneity and complexity for personalized treatments, as well as for enhancing survival predictions. Despite the progress made in integrating pathology and genomic data, most existing methods cannot mine the complex inter-modality relations thoroughly. Additionally, identifying explainable features from these models that govern preclinical discovery and clinical prediction is crucial for cancer diagnosis, prognosis, and therapeutic response studies. We propose PONET- a novel biological pathway informed pathology-genomic deep model that integrates pathological images and genomic data not only to improve survival prediction but also to identify genes and pathways that cause different survival rates in patients. Empirical results on six of The Cancer Genome Atlas (TCGA) datasets show that our proposed method achieves superior predictive performance and reveals meaningful biological interpretations. The proposed method establishes insight on how to train biological informed deep networks on multimodal biomedical data which will have general applicability for understanding diseases and predicting response and resistance to treatment.

## 1 Introduction

Manual examination of haematoxylin and eosin (H&E)-stained slides of tumour tissue by pathologists is currently the state-of-the-art for cancer diagnosis (Chan, 2014). The recent advancements in deep learning for digital pathology have enabled the use of whole-slide images (WSI) for computational image analysis tasks, such as cellular segmentation (Pan et al., 2017; Hou et al., 2020), tissue classification and characterisation (Hou et al., 2016; Hekler et al., 2019; Iizuka et al., 2020). While H&E slides are important and sufficient to establish a profound diagnosis, genomics data can provide a deep characterisation of the tumour on the molecular level potentially offering the chance for prognostic and predictive biomarker discovery.

Cancer prognosis via survival outcome prediction is a standard method used for biomarker discovery, stratification of patients into distinct treatment groups, and therapeutic response prediction (Cheng et al., 2017; Ning et al., 2020). WSIs exhibit enormous heterogeneity and can be as large as $150,000 \times 150,000$ pixels. Most approaches adopt a two-stage multiple instance learning-based (MIL) approach for representation learning of WSIs, in which: 1) instance-level feature representations are extracted from image patches in the WSI, and then 2) global aggregation schemes are applied to the bag of instances to obtain a WSI-level representation for subsequent supervision (Hou et al., 2016; Courtiol et al., 2019; Wulczyn et al., 2020; Lu et al., 2021). Therefore, multimodal survival prediction faces an additional challenge due to the large data heterogeneity gap between WSIs and genomics, and many existing approaches use simple multimodal fusion mechanisms for feature integration, which prevents mining important multimodal interactions (Mobadersany et al., 2018; Chen et al., 2022b;a).

The incorporation of biological pathway databases in a model takes advantage of leveraging prior biological knowledge so that potential prognostic factors of well-known biological functionality can be identified (Hao et al., 2018). Moreover, encoding biological pathway information into the neural networks achieved superior predictive performance compared with established models (Elmarakeby et al., 2021).

Based on the current challenges in multimodal fusion of pathology and genomics and the potential prognostic interpretation to link pathways and clinical outcomes in pathway-based analysis, we

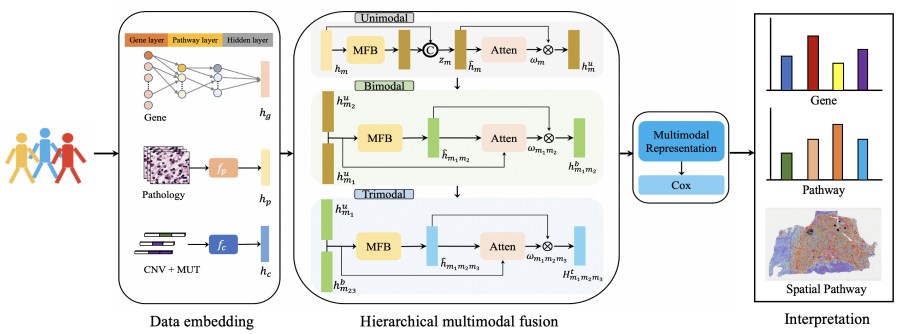

Figure 1: Overview of PONET model.

propose a novel biological pathway informed pathology-genomic deep model, PONET, that uses H&E WSIs and genomic profile features for survival prediction. The proposed method contains four major contributions: 1) PONET formulates a biological pathway informed deep hierarchical multimodal integration framework for pathological images and genomic data; 2) PONET captures diverse and comprehensive modality-specific and cross modality relations among different data sources based on factorized bilinear model and graph fusion network; 3) PONET reveals meaningful model interpretations on both genes and pathways for potential biomarker and therapeutic target discovery; PONET also shows spatial visualization of the top genes/pathways which has enormous potential for novel and prognostic morphological determinants; 4) We evaluate PONET on six public TCGA datasets which showed superior survival prediction comparing to state-of-the-art methods. Fig. 1 shows our model framework.

## 2 RELATED WORK

**Multimodal Fusion.** Earlier works on multimodal fusion focus on early fusion and late fusion. Early fusion approaches fuse features by simple concatenation which cannot fully explore intra-modality dynamics (Wöllmer et al., 2013; Poria et al., 2016; Zadeh et al., 2016). In contrast, late fusion fuses different modalities by weighted averaging which fails to model cross-modal interactions (Nojavanasghari et al., 2016; Kampman et al., 2018). The exploitation of relations within each modality has been successfully introduced in cancer prognosis via bilinear model (Wang et al., 2021b) and graph-based model (Subramanian et al., 2021). Adversarial Representation Graph Fusion (ARGF) (Mai et al., 2020) interprets multimodal fusion as a hierarchical interaction learning procedure where firstly bimodal interactions are generated based on unimodal dynamics, and then trimodal dynamics are generated based on bimodal and unimodal dynamics. We propose a new hierarchical fusion framework with modality-specific and cross-modality attentional factorized bilinear modules to mine the comprehensive modality interactions. Our proposed hierarchical fusion framework is different from ARGF in the following ways: 1) We take the sum of the weighted modality-specific representation as the unimodal representation instead of calculating the weighted average of the modality-specific representation in ARGF; 2) For higher level's fusion, ARGF takes the original embeddings of each modality as input while we use the weighted modality-specific representations; 3) We argue that ARGF takes redundant information during their trimodal dynamics.

**Multimodal Survival Analysis.** There have been exciting attempts on multimodal fusion of pathology and genomic data for cancer survival prediction (Mobadersany et al., 2018; Cheerla & Gevaert, 2019; Wang et al., 2020). However, these multimodal fusion based methods fail to explicitly model the interaction between each subset of multiple modalities. Kronecker product considers pairwise interactions of two input feature vectors by producing a high-dimensional feature of quadratic expansion (Zadeh et al., 2017), and showed its superiority in cancer survival prediction (Wang et al., 2021b; Chen et al., 2022b;a). Despite of promising results, using Kronecker product in multimodal fusion may introduce a large number of parameters that may lead to high computational cost and a risk of overfitting (Kim et al., 2017; Liu et al., 2021), thus limiting its applicability and improvement in performance. To overcome this drawback, hierarchical factorized bilinear fusion for cancer survival prediction (HFBSurv) (Li et al., 2022) uses factorized bilinear model to fuse genomic and image features which dramatically reduces computational complexity. PONET differs from HFBSurv in two ways: 1) PONET's multimodal framework has three levels of hierarchical fusion module

including unimodal, bimodal, and trimodal fusion while HFBSurv only considers within-modality and cross-modality fusion which we argue it is not adequate for mining the comprehensive interactions; 2) PONET leverages biological pathway informed network for better prediction and meaningful interpretation purposes.

**Pathway-associated Sparse Neural Network.** Pathway-based analysis is an approach that a number of studies have been investigated to improve both predictive performance and biological interpretability (Jin et al., 2014; Cirillo et al., 2017; Hao et al., 2018; Elmarakeby et al., 2021). Moreover, pathway-based approaches have shown more reproducible analysis results than gene expression data analysis alone (Li et al., 2015; Mallavarapu et al., 2017). These pathway-based deep neural networks can only model genomic data which severely inhibits their applicability in current biomedical research. Additionally, the existing pathway-associated sparse neural network structures are limited for disease mechanism investigation: there is only one pathway layer in PASNet (Hao et al., 2018) which contains limited biological prior information to deep dive into the hierarchical pathway and biological process relationships; P-NET (Elmarakeby et al., 2021) calculates the final prediction by taking the average of all the gene and pathways layers' outputs, and this will bias the learning process because it will put more weights for some layers' outputs while underestimates the others.

## 3 METHODOLOGY

### 3.1 PROBLEM FORMULATION AND NOTATIONS

The model architecture of PONET is presented in Fig. 1, where three modalities are included as input: gene expression $g \in \mathbb{R}^{d_g}$, pathological image $p \in \mathbb{R}^{d_p}$, and copy number (CNV) + mutation (MUT) $CNV + MUT \in \mathbb{R}^{d_c}$, with $d_p$ being the dimensionality of $p$ and so on. We define a hierarchical factorized bilinear fusion model for PONET. We build a sparse biological pathway-informed embedding network for gene expression, and a fully connected (FC) embedding layer for both preprocessed pathological image feature ($f_p$) and the copy number + mutation ($f_c$) to map feature into similar embedding space for alleviating the statistical property differences between modalities, the three network architecture details are in Appendix C.1. We label the three modality embeddings as $h_m$, $m \in \{g, p, c\}$, the superscript/subscript $u$, $b$, and $t$ represents unimodal fusion, biomodal fusion and trimodal fusion. After that, the embeddings of each modality is first used as input for unimodal fusion to generate the modality-specific representation $h_m^u = \omega^m \hat{h}_m$, $\omega^m$ represents the modality-specific importance, the feature vector of the unimodal fusion is the sum of all modality-specific representations $h_u = \sum_m h_m^u$. In the bimodal fusion, modality-specific representations from the output of unimodal fusion are fused to yield cross-modality representations $h_{m_1 m_2}^b = \omega_{m_1 m_2} \hat{h}_{m_1 m_2}$, $m_1, m_2 \in \{p, c, g\}$ and $m_1 \neq m_2$, $\omega_{m_1 m_2}$ represents the corresponding cross-modality importance. Similarly, the feature vector of bimodal fusion is calculated as $h_b = \sum_{m_1, m_2} h_{m_1 m_2}^b$. We propose to build a trimodal fusion to take each cross-modality representation from the output of bimodal fusion to thoroughly mine the interactions. Simiarly as the bimodal fusion architecture, trimodal fusion feature vector will be $h_t = \sum_{m_1, m_2, m_3} \omega_{m_1 m_2 m_3} \hat{h}_{m_1 m_2 m_3}$, $m_1, m_2, m_3 \in \{p, c, g\}$ and $m_1 \neq m_2 \neq m_3$, $\omega_{m_1 m_2 m_3}$ represents the corresponding trimodal importance. Finally, PONET concatenates $h_u$, $h_b$, $h_t$ to obtain the final comprehensive multimodal representation and pass it to the Cox porportional harzard model (Cox, 1972; Cheerla & Gevaert, 2019) for survival prediction. In the following sections we will describe our hierarchical factorized bilinear fusion framework, $l$, $o$, $s$ represents the dimensionality of $h_m$, $z_m$, $\hat{h}_{m1m2}$.

### 3.2 SPARSE NETWORK

We design the sparse gene-pathway network consisting of one gene layer followed by three pathway layers. A patient sample of $e$ gene expressions is formed as a column vector, which is denoted by $\mathbf{X} = [x_1, x_2, ..., x_e]$, each node represents one gene. The gene layer is restricted to have connections reflecting the gene-pathway relationships curated by the Reactome pathway dataset (Fabregat et al., 2020). The connections are encoded by a binary matrix $\mathbf{M} \in \mathbb{R}^{a \times e}$, where $a$ is number of pathways and $e$ is number of genes, an element of $\mathbf{M}$, $m_{ij}$, is set to one if gene $j$ belongs to pathway $i$. The connections that do not exist in the Reactome pathway dataset will be zero-out. For the next pathway-pathway layers, a similar scheme is applied to control the connection between consecutive layers to reflect the parent-child hierarchical relationships that exist in the Reactome dataset. The

Figure 2: Overall framework of the visual representation extraction using pre-trained self-supervised vision transformer.

output of each layer is calculated as

$$y = f[(\mathbf{M} * \mathbf{W})^T \mathbf{X} + \epsilon] \tag{1}$$

where $f$ is the activation function, $\mathbf{M}$ represents the binary matrix, $\mathbf{W}$ is the weights matrix, $\mathbf{X}$ is the input matrix, $\epsilon$ is the bias vector, and $*$ is the Hadamard product. We use tanh for the activation of each node. We allow the information flow from the biological prior informed network starting from the first gene layer to the last pathway layer, and we label the last layer output embeddings of the sparse network for gene expression as $h_g$.

### 3.3 UNIMODAL FUSION

Bilinear models (Tenenbaum & Freeman, 2000) provide richer representations than linear models. Given two feature vectors in different modalities, e.g., the visual features $x \in \mathbb{R}^{m \times 1}$ for an image and the genomic features $y \in \mathbb{R}^{n \times 1}$ for a genomic profile, bilinear model uses a quadratic expansion of linear transformation considering every pair of features:

$$z_i = x^T W_i y \tag{2}$$

where $W_i \in \mathbb{R}^{m \times n}$ is a projection matrix, $z_i \in \mathbb{R}$ is the output of the bilinear model. Bilinear models introduce a large number of parameters which potentially lead to high computational cost and overfitting risk. To address these issues, Yu et al. (2017) develop the Multi-modal Factorized Bilinear pooling (MFB) method, which enjoys the dual benefits of compact output features and robust expressive capacity.

Inspired by the MFB (Yu et al., 2017) and its application in pathology and genomic multimodal learning (Li et al., 2022), we propose unimodal fusion to capture modality-specific representations and quantify their importance. The unimodal fusion takes the embedding of each modality $h_m$ as input and factorizes the projection matrix $W_i$ in Eq. (2) as two low-rank matrices:

$$\begin{aligned} z_i &= h_m^T W_i h_m = \sum_{d=1}^{k} h_m^T u_{m,d} v_{m,d}^T h_m \\ &= 1^T (U_{m,i}^T h_m \circ V_{m,i}^T h_m), m \in \{p, c, g\} \end{aligned} \tag{3}$$

we get the output feature $z_m$:

$$z_m = \text{SumPooling}\left(\tilde{U}_m^T h_m \circ \tilde{V}_m^T h_m, k\right), m \in \{p, c, g\} \tag{4}$$

where $k$ is the latent dimensionality of the factorized matrices. SumPooling $(x, k)$ function performs sum pooling over $x$ by using a 1-D non-overlapped window with the size k, $\tilde{U}_m \in \mathbb{R}^{l \times ko}$ and $\tilde{V}_m \in \mathbb{R}^{l \times ko}$ are 2-D matrices reshaped from $U_m$ and $V_m$, $U_m = [U_{m,1}, \dots, U_{m,h}] \in \mathbb{R}^{l \times k \times o}$ and $V_m = [V_{m,1}, \dots, V_{m,h}] \in \mathbb{R}^{l \times k \times o}$. Each modality-specific representation $\hat{h}_m \in \mathbb{R}^{l+o}$ is obtained as:

$$\hat{h}_m = h_m \copyright z_m, m \in \{p, c, g\} \tag{5}$$

where $\copyright$ denotes vector concatenation. We also introduce an modality attention network $Atten \in \mathbb{R}^{l+o} \to \mathbb{R}^1$ to determine the weight for each modality-specific representation to quantify its importance:

$$\omega_m = Atten(\hat{h}_m; \Theta_{Atten}), m \in \{p, c, g\} \tag{6}$$

where $\omega_m$ is the weight of modality $m$. In practice, $Atten$ consists a sigmoid activated dense layer parameterized by $\Theta_{Atten}$. Therefore, the output of each modality in unimodal fusion, $h_m^u$, is denoted

as $\omega_m \hat{h}_m \in \mathbb{R}^{l+o}, m \in \{p, c, g\}$. Accordingly, the output of unimodal fusion, $h_u$, is the sum of each weighted modality-specific representation $\omega_m \hat{h}_m, m \in \{p, c, g\}$ which is different from ARGF (Mai et al., 2020) that used the weighted average of different modalities as the unimodal fusion output.

### 3.4 BIMODAL AND TRIMODAL FUSION

The goal of bimodal fusion is to fuse diverse information of different modalities and quantify different importance for them. After receiving the modality-specific representations $h_m^u$ from the unimodal fusion, we can generate the cross-modality representation $\hat{h}_{m_1 m_2} \in \mathbb{R}^s$ similar to Eq. (4) :

$$\hat{h}_{m1,m2} = \text{Sum Pooling} \left( \tilde{U}_{m_1}^T h_{m_1}^u \circ \tilde{V}_{m_2}^T h_{m_2}^u, k \right),$$
$$m_1, m_2 \in \{p, c, g\}, m_1 \neq m_2 \tag{7}$$

where $\tilde{U}_{m_1}^T \in \mathbb{R}^{(l+o) \times ks}$ and $\tilde{V}_{m_2}^T \in \mathbb{R}^{(l+o) \times ks}$ are 2-D matrices reshaped from $U_{m_1}$ and $V_{m_2}$ and $U_{m_1} = [U_{m_1,1}, \ldots, U_{m_1,s}] \in \mathbb{R}^{(l+o) \times k \times s}$ and $V_{m_2} = [V_{m_2,1}, \ldots, V_{m_2,s}] \in \mathbb{R}^{(l+o) \times k \times s}$. We leverage a bimodal attention network (Mai et al., 2020) to identify the importance of the cross-modality representation. The similarity $S_{m_1 m_2} \in \mathbb{R}^1$ of $h_{m_1}^u$ and $h_{m_2}^u$ is first estimated as follows:

$$S_{m_1,m_2} = \sum_{i=1}^{l+o} \left( \frac{e^{\omega_{m_1} h_{m_1,i}^u}}{\sum_{j=1}^{l+o} e^{\omega_{m_1} h_{m_1,j}^u}} \right) \left( \frac{e^{\omega_{m_2} h_{m_2,i}^u}}{\sum_{j=1}^{l+o} e^{\omega_{m_2} h_{m_2,j}^u}} \right) \tag{8}$$

where the computed similarity is in the range of 0 to 1. Then, the cross-modality importance $\omega_{m_1 m_2}$ is obtained by:

$$\omega_{m_1 m_2} = \frac{e^{\hat{\omega}_{m_i m_j}}}{\sum_{m_i \neq m_j} e^{\hat{\omega}_{m_i m_j}}}, \hat{\omega}_{m_1 m_2} = \frac{\omega_{m_1} + \omega_{m_2}}{S_{m_1 m_2} + S_0} \tag{9}$$

where $S_0$ represents a pre-defined term controlling the relative contribution of similarity and modality-specific importance, and here is set to 0.5. Therefore, the output of bimodal fusion, $h_b$, is the sum of each weighted cross-modality representation $\omega_{m_1 m_2} \hat{h}_{m_1 m_2}, m_1, m_2 \in \{p, c, g\}$ and $m_1 \neq m_2$.

In the trimodal fusion, each bimodal fusion output is fused with the unimodal fusion output that does not contribute to the formation of the bimodal fusion. The output for each corresponding trimodal representation is $\hat{h}_{m_1 m_2 m_3}$. In addition, a trimodal attention was applied to identify the importance of each trimodal representation, $\omega_{m_1 m_2 m_3}$. The output of the trimodal fusion, $h_t$, is the sum of each weighted trimodal representation $\omega_{m_1 m_2 m_3} \hat{h}_{m_1 m_2 m_3}, m_1, m_2, m_3 \in \{p, c, g\}$ and $m_1 \neq m_2 \neq m_3$.

### 3.5 SURVIVAL LOSS FUNCTION

We train the model through the Cox partial likelihood loss (Cheerla & Gevaert, 2019) with $l_1$ regularization for survival prediction, which is defined as:

$$\ell(\Theta) = - \sum_{i:E_i=1} \left( \hat{\mathfrak{h}}_\Theta (x_i) - \log \sum_{j:T_i > T_j} \exp \left( \hat{\mathfrak{h}}_\Theta (x_j) \right) \right) + \lambda \left( \|\Theta\|_1 \right) \tag{10}$$

where the values $E_i, T_i$ and $x_i$ for each patient represent the survival status, the survival time and the feature, respectively. $E_i = 1$ means event while $E_i = 0$ represents censor. $\hat{\mathfrak{h}}_\Theta$ is the neural network model trained for predicting the risk of survival, $\Theta$ is the neural network model parameters, and $\lambda$ is a regularization hyperparameter to avoid overfitting.

## 4 EXPERIMENTS

### 4.1 EXPERIMENTAL SETUP

**Datasets.** To validate our proposed method, we used six cancer datasets from The Cancer Genome Atlas (TCGA), a public cancer data consortium that contains matched diagnostic WSIs and genomic

data with labeled survival times and censorship statuses. The genomic profile features (mutation status, copy number variation, RNA-Seq expression) are preprocessed by Porpoise [1] (Chen et al., 2022b). For this study, we used the following cancer types: Bladder Urothelial Carcinoma (BLCA) (n = 437), Kidney Renal Clear Cell Carcinoma (KIRC) (n = 350), Kidney Renal Papillary Cell Carcinoma (KIRP) (n = 284), Lung Adenocarcinoma (LUAD) (n = 515), Lung Squamous Cell Carcinoma (LUSC) (n = 484), Pancreatic adenocarcinoma (PAAD) (n = 180). We downloaded the same diagnostic WSIs from TCGA website [2] that used in Porpoise study to match the paired genomic features and survival times. The feature alignment table for all the cancer type is in Appendix A. For each WSI, automated segmentation of tissue was performed. Following segmentation, image patches of size $224 \times 224$ were extracted without overlap at the 20 X equivalent pyramid level from all tissue regions identified while excluding the white background and selecting only patches with at least 50% tissue regions. Subsequently, visual representation of those patches are extracted with a vision transformer (Wang et al., 2021a) pre-trained on the TCGA dataset through a self-supervised constructive learning approach, such that each patch is represented as a $1 \times 2048$ vector. Fig. 2 shows the framework for the visual representation extraction by vision transformer (VIT). Survival outcome information is available at the patient-level, we aggregated the patch level feature into slide level feature representations based on attention-based method (Lu et al., 2021; Ilse et al., 2018), please check the algorithm details in Appendix B.4.

**Baselines.** Using the same 5-fold cross-validation splits for evaluating PONET, we implemented and evaluated six state-of-the-art methods for survival outcome prediction. Additionally, we included three variations of PONET: a) PONET-O represents only genomic data and pathway architecture for the gene expression are included in the model; b) PONET-OH represents only genomic and pathological image data but without pathway architecture in the model; c) PONET is our full model. For all methods, we use the same VIT feature extraction pipeline for WSIs, as well as identical training hyperparameters and loss function for supervision. Training details and the parameters tuning can be found in the Appendix C.2.

**CoxPH** (Cox, 1972) represents the standard Cox proportional hazard models (Appendix B.1).
**DeepSurv** (Katzman et al., 2018) is the deep neural network version of CoxPH model.
**Pathomic Fusion** (Chen et al., 2022a) as a pioneered deep-learning based framework for predicting survival outcome by fusing pathology and genomic multimodal data, in which Kronecker product is taken to model pairwise feature interactions across modalities.
**GPDBN** (Wang et al., 2021b) adopts Kronecker product to model inter-modality and intra-modality relations between pathology and genomic data for cancer prognosis prediction.
**HFBSurv** (Li et al., 2022) extended GPDBN using factorized bilinear model to fuse genomic and pathology features in a within-modality and cross-modalities hierarchical fusion.
**Porpoise** (Chen et al., 2022b) applied discrete survival model and Kronecker product to fuse pathology and genomic data for survival prediction (Zadeh & Schmid, 2020).

**Evaluation.** For each cancer dataset, we used the cross-validated concordance index (C-Index) (Appendix B.2) (Harrell et al., 1982) to measure the predictive performance of correctly ranking the predicted patient risk scores with respect to overall survival.

## 4.2 RESULTS

**Comparison with Baselines.** In combing pathology image, genomics, and pathway network via PONET, our approach outperforms CoxPH models, unimodal networks, and previous deep learning based approaches on pathology-genomic-based survival outcome prediction (Table 1). From the results, deep learning-based approaches generally exhibit better performance than CoxPH model. PONET achieves superior C-index value in all six cancer types. All versions of PONET outperform Pathomic Fusion by a big margin. Pathomic Fusion uses Kronecker product to fuse the two modalities and that's also the reason why other advanced fusion methods, like GPDBN and HFBSurv, achieves better performance. Also, we argue that Pathomic Fusion extracts the region of interest of pathology image for feature extraction might limit the understanding of the tumor microenvironment of the whole slide. HFBSurv shows better performance than GPDBN and Pathomic Fusion which is consistent with their findings, and these results further demonstrate that the hierarchical factorized

---

[1]https://github.com/mahmoodlab/PORPOISE
[2]https://www.cancer.gov/about-nci/organization/ccg/research/structural-genomics/tcga

Table 1: C-Index (mean ± standard deviation) of PONET and ablation experiments in TCGA survival prediction. Top two performers are highlighted in bold.

| Model | TCGA-BLCA | TCGA-KIRC | TCGA-KIRP | TCGA-LUAD | TCGA-LUSC | TCGA-PAAD |
|---|---|---|---|---|---|---|
| CoxPH (Age + Gender) (Cox, 1972) | 0.525 ± 0.130 | 0.550 ± 0.070 | 0.544 ± 0.050 | 0.531 ± 0.082 | 0.532 ± 0.094 | 0.539 ± 0.092 |
| DeepSurv (Kampman et al., 2018) | 0.580 ± 0.062 | 0.620 ± 0.043 | 0.560 ± 0.063 | 0.534 ± 0.077 | 0.541 ± 0.066 | 0.544 ± 0.076 |
| GPDBN (Wang et al., 2021b) | 0.612 ± 0.042 | 0.647 ± 0.073 | 0.669 ± 0.109 | 0.565 ± 0.057 | 0.545 ± 0.063 | 0.571 ± 0.060 |
| HFBSurv (Li et al., 2022) | 0.622 ± 0.043 | 0.667 ± 0.053 | 0.769 ± 0.109 | 0.581 ± 0.017 | 0.548 ± 0.049 | 0.591 ± 0.052 |
| Pathomic Fusion (Chen et al., 2022a) | 0.586 ± 0.062 | 0.598 ± 0.060 | 0.577 ± 0.032 | 0.543 ± 0.065 | 0.523 ± 0.045 | 0.545 ± 0.064 |
| Porpoise (Chen et al., 2022b) | 0.617 ± 0.048 | **0.711 ± 0.051** | **0.811 ± 0.089** | 0.586 ± 0.056 | 0.527 ± 0.043 | 0.591 ± 0.064 |
| **PONET-O (ours)** | 0.596 ± 0.056 | 0.664 ± 0.056 | 0.761 ± 0.093 | **0.623 ± 0.062** | 0.538 ± 0.037 | **0.598 ± 0.027** |
| **PONET-OH (ours)** | **0.625 ± 0.063** | 0.695 ± 0.043 | 0.776 ± 0.123 | 0.618 ± 0.049 | **0.553 ± 0.049** | 0.591 ± 0.050 |
| **PONET (ours)** | **0.643 ± 0.037** | **0.726 ± 0.056** | **0.829 ± 0.054** | **0.646 ± 0.047** | **0.567 ± 0.066** | **0.639 ± 0.080** |

Table 2: Evaluation of PONET on different fusion methods and pathway designs by C-index (mean ± standard deviation). Best performer is highlighted in bold.

| | Methods | TCGA-BLCA | TCGA-KIRP | TCGA-LUAD | TCGA-LUSC | TCGA-PAAD |
|---|---|---|---|---|---|---|
| Single fusion | Simple concatenation | 0.585 ± 0.045 | 0.652 ± 0.049 | 0.554 ± 0.065 | 0.525 ± 0.066 | 0.568 ± 0.075 |
| | Element-wise addition | 0.592 ± 0.047 | 0.655 ± 0.055 | 0.587 ± 0.065 | 0.522 ± 0.046 | 0.588 ± 0.055 |
| | Tensor fusion (Zadeh et al., 2017) | 0.605 ± 0.046 | 0.775 ± 0.053 | 0.595 ± 0.060 | 0.545 ± 0.045 | 0.592 ± 0.061 |
| Hierarchical fusion | Unimodal | 0.596 ± 0.035 | 0.783 ± 0.063 | 0.611 ± 0.056 | 0.553 ± 0.073 | 0.595 ± 0.053 |
| | Bimodal | 0.602 ± 0.062 | 0.789 ± 0.053 | 0.601 ± 0.056 | 0.552 ± 0.051 | 0.598 ± 0.083 |
| | ARGF (Mai et al., 2020) | 0.597 ± 0.054 | 0.792 ± 0.054 | 0.614 ± 0.051 | 0.556 ± 0.063 | 0.602 ± 0.065 |
| | Unimodal + Bimodal | 0.614 ± 0.052 | 0.803 ± 0.061 | 0.631 ± 0.044 | **0.578 ± 0.058** | 0.615 ± 0.057 |
| Pathway design | PASNet (Hao et al., 2018) | 0.606 ± 0.045 | 0.793 ± 0.051 | 0.621 ± 0.061 | 0.551 ± 0.069 | 0.625 ± 0.057 |
| | P-NET (Elmarakeby et al., 2021) | 0.622 ± 0.047 | 0.802 ± 0.071 | 0.625 ± 0.045 | 0.562 ± 0.054 | 0.627 ± 0.073 |
| | PONET | **0.643 ± 0.037** | **0.829 ± 0.054** | **0.641 ± 0.046** | 0.567 ± 0.066 | **0.639 ± 0.070** |

bilinear model can better mine the rich complementary information among different modalities compare to Kronecker product. Porpoise performs similarly with PONET on TCGA-KIRC and TCGA-KIRP and outperformed HFBSurv in these two studies, this probably is due to Porpoise partitioned the survival time into different non-overlapping bins and parameterized it as a discrete survival model (Zadeh & Schmid, 2020) which works better for these two cancer types. In other cases, Porpoise performs similarly with HFBSurv. Note: the results of Porpoise are from their paper (Chen et al., 2022b).

Additionally, we can see that PONET consistently outperforms PONET-O and PONET-OH indicating that the effectiveness of the biological pathway-informed neural network and the contribution of pathological image for the overall survival prediction.

**Ablation Studies.** To assess the impact of hierarchical factorized bilinear fusion strategy is indeed effective, we compare PONET with four single-fusion methods: 1) Simple concatenation: concatenate each modality embeddings; 2) Element-wise addition: element-wise addition from each modality embeddings; 3) Tensor fusion (Zadeh et al., 2017): Kronecker product from each modality embeddings. Table 2 shows the C-index values of different methods. We can see that PONET achieves the best performance and shows remarkable improvement over single-fusion methods on different cancer type datasets. For example, PONET outperforms the Simple concatenation by 8.4% (TCGA-BLCA), 27% (TCGA-KIRP), 15% (TCGA-LUAD), 8.0% (TCGA-LUSC), and 11.4% (TCGA-PAAD), etc.

Furthermore, we adopted five different configurations of PONET to evaluate each hierarchical component of the proposed method: 1) Unimodal: unimodal fusion output as the final feature representation; 2) Bimodal: bimodal fusion output as the final feature representation; 3) Unimodal + Bimodal: hierarchical (include both unimodal and bimodal feature representation) fusion; 4) ARGF: ARGF(Mai et al., 2020) fusion strategy; 5) PONET: our proposed hierarchical strategy by incorporating unimodal, bimodal and trimodal fusion output. As shown in Table 2, Unimodal + Bimodal performs better than Unimodal and Bimodal which demonstrates that Unimodal + Bimodal can capture the relations within each modality and across modalities. ARGF performs worse than Unimodal + Bimodal and far worse than PONET across all the cancer types. PONET outperforms Unimodal + Bimodal in 4 out of 5 cancer types indicating that three layers of hierarchical fusion can mine the comprehensive interactions among different modalities.

To evaluate our sparse gene-pathway network design, we compare PONET with PASNet (Hao et al., 2018) and P-NET (Elmarakeby et al., 2021) pathway architecture, PASNet performs the worst due to the fact that it only has one pathway layer in the network, and thus limited prior information was used to predict the outcome. PONET constantly outperforms P-NET across all the cancer types, which

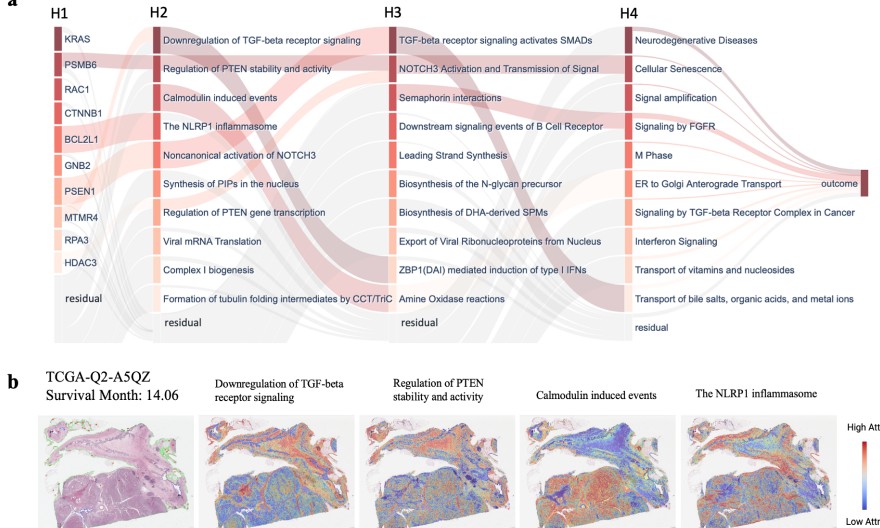

Figure 3: Inspecting and interpreting PONET on TCGA-KIRP. **a**: Sankey diagram visualization of inner layers of PONET shows the estimated relative importance of different nodes in each layer. Nodes in the first layer represent genes; the next layers represent pathways; and the final layer represents the model outcome. Different layers are linked by weights. Nodes with darker colours are more important, while transparent nodes represent the residual importance of undisplayed nodes in each layer, H1 presents gene layer and H2-H4 represent pathway layers; **b**: Co-attention visualization of top 4 ranked pathways in one case of TCGA-KIRP.

demonstrates that averaging all the intermediate layers' output for the final prediction cannot fully capture the prior information flow among the biological hierarchical structures.

**Model Interpretation.** We discuss the model interpretation results for cancer type TCGA-KIRP here and the results for other cancer types are included in the Appendix C.3. To understand the interactions between different genes, pathways and biological processes that contributed to the predictive performance and to study the paths of impact from the input to the outcome, we visualized the whole structure of PONET with the fully interpretable layers after training (Fig. 3 a). To evaluate the relative importance of specific genes contributing to the model prediction, we inspected the genes layer and used the Integrated Gradients attribution (Sundararajan et al., 2017) method to obtain the total importance score of genes, and the modified ranking algorithm details are included in the Appendix B.5. Highly ranked genes included KRAS, PSMB6, RAC1, and CTNNB1 which are known kidney cancer drivers previously (Yang et al., 2017; Shan et al., 2017; Al-Obaidy et al., 2020; Guo et al., 2022). GBN2, a member of the guanine nucleotide-binding proteins family, which has been reported that the decrease of its expression reduced tumor cell proliferation (Zhang et al., 2019). A recent study identified strong dependency on BCL2L1, which encodes the BCL-XL anti-apoptotic protein, in a subset of kidney cancer cells (Grubb et al., 2022). This biological interpretability revealed established and novel molecular features contributing kidney cancer. In addition, PONET selected a hierarchy of pathways relevant to the model prediction, including down regulation of TGF-$\beta$ receptor signaling, regulation of PTEN stability and activity, the NLRP1 inflammasome, and noncanonical activation of NOTCH3 by PSEN1, PSMB6, and BCL2L1. TGF-$\beta$ signaling is increasingly recognized as a key driver in cancer, and in progressive cancer tissues TGF-$\beta$ promotes tumor formation, and its increased expression often correlates with cancer malignancy (Han et al., 2018). Noncanonical activation of NOTCH3 was reported to limit tumour angiogenesis and plays vital roles in kidney disease (Lin et al., 2017).

To further inspect the pathway spatial association with WSI slide we adopted the co-attention survival method MCAT (Chen et al., 2021) between WSIs and genomic features on the top pathways of the second layer, visualized as a WSI-level attention heatmap for each pathway genomic embedding in Fig. 3 b (algorithm details are included in the Appendix B.6). We used the gene list from top 4 pathways as the genomic features and trained MCAT on TCGA-KIRP dataset for survival prediction. Overall, we observe that high attention in different pathways showed different spatial pattern associations with the slide. This heatmap can reflect genotype-phenotype relationships in cancer pathology. The high attention regions (red) of different pathways in the heatmap have positive associations with the pre-

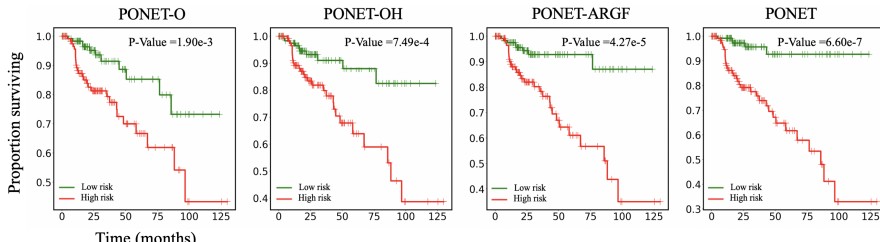

Figure 4: Kaplan-Meier analysis of patient stratification of low and high risk patients via four variatons of PONET on TCGA-KIRP. Low and high risks are defined by the median 50% percentile of hazard predictions via each model prediction. Logrank test was used to test for statistical significance in survival distributions between low and high risk patients.

dicted death risk while the low attention regions (blue) have negative associations with the predicted risk. By further check the cell types in high attention patches we can gain insights of prognostic morphological determinants and have a better understanding about the complex tumor microenvironment.

**Patient Stratification.** In visualizing the Kaplan-Meier survival curves of predicted high risk and low risk patient populations, we plot four variations of PONET in Fig. 4. PONET-ARGF represents the model that we use the hierarchical fusion strategy of ARGF in our pathway-informed PONET model. From the results, PONET enables easy separation of patients into low and high risk groups with remarkably better stratification (P-Value=6.60e-7) in comparison to the others.

Table 3: Comparison of model complexity

| Methods | Number of Parameters | FLOPS |
|---|---|---|
| Pathomic Fusion | 175M | 168G |
| GPDBN | 82M | 91G |
| HFBSurv | 0.3M | 0.5G |
| PONET | 2.8M | 3.1G |

**Complexity Comparison.** We compared PONET with Pathomic Fusion, GPDBN, and HFBSurv since both Pathomic Fusion and GPDBN are based on Kronecker product to fuse different modalities while GPDBN and HFBSurv modeled inter-modality and intra-modality relations which have similar consideration to our method. As illustrated in Table 3, PONET has 2.8M (M = Million) trainable parameters, which is approximately 1.6%, 3.4%, and 900% of the number of parameters of Pathomic Fusion, GPDBN, and HFBSurv. To assess the time complexity of PONET and the competitive methods, we calculate floating-point operations per second (FLOPS) of each method in testing. The results in Table 3 show that PONET needs 3.1G during testing, compared with 168G, 91G, and 0.5G in Pathomic Fusion, GPDBN, and HFBSurv. The main reason for fewer trainable parameters and number of FLOPS lies in that PONET and HFBSurv performs multimodal fusion using factorized bilinear model, and can significantly reduce the computational complexity and meanwhile obtain more favorable performance. PONET has one additional trimodal fusion which explains why it has more trainable parameters than HFBSurv.

## 5 CONCLUSION

In this study, we pioneer propose a novel biological pathway-informed hierarchical multimodal fusion model that integrates pathology image and genomic profile data for cancer prognosis. In comparison to previous works, PONET deeply mines the interaction from multimodal data by conducting unimodal, biomodal and trimodal fusion step by step. Empirically, PONET demonstrates the effectiveness of the model architecture and the pathway informed network for superior predictive performance. Specifically, PONET provides insight on how to train biological informed deep networks on multimodal biomedical data for biological discovery in clinic genomic contexts which will be useful for other problems in medicine that seek to combine heterogeneous data streams for understanding diseases and predicting response and resistance to treatment.

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

Table 4: TCGA Data Feature Alignment Summary

| Cancer Type | WSI | CNV | MUT | RNA | WSI+CNV+MUT | WSI+MUT+RNA | ALL |
|---|---|---|---|---|---|---|---|
| BLCA | 454 | 443 | 452 | 450 | 441 | 448 | **437** |
| KIRC | 517 | 509 | 357 | 514 | 352 | 355 | **350** |
| KIRP | 294 | 291 | 286 | 293 | 284 | 285 | **284** |
| LUAD | 528 | 522 | 523 | 522 | 519 | 519 | **515** |
| LUSC | 505 | 502 | 489 | 503 | 486 | 487 | **484** |
| PAAD | 208 | 201 | 187 | 195 | 187 | 180 | **180** |

## A  DATA

Table 3 in Appendix A shows the number of patients with matched different data modalities: WSI (Whole slide image), CNV (Copy number), MUT (Muation), RNA (RNA-Seq gene expression). For each TCGA dataset and each patient we have preprocessed data dimensions $\mathbf{d}_{\mathrm{g}} \in \mathbb{R}^{1 \times 2000}$ (RNA), $\mathbf{d}_{\mathrm{c}} \in \mathbb{R}^{1 \times 227}$ (CNV + MUT), and $\mathbf{d}_{\mathrm{p}} \in \mathbb{R}^{1 \times 32}$ (WSI) which will be used for our multimodal fusion.

## B  METHODS

### B.1  COX PROPORTIONAL HAZARD MODEL

In survival analysis, we are interested in modeling the continuous time $T$ until some event of interest (i.e. survival time). The survival function $S(t) = \mathbb{P}(T > t_0) = 1 - \int_0^t f(s)ds$ is the probability of a individual surviving longer than time $t_0$, where $f$ is the probability density function of survival times. We denote the probability that an event occurs in an infinitesimal interval after time $t$, given it has not yet occurred at time $t$ as the hazard function $\lambda(t)$:

$$\lambda(t) = \lim_{\delta \to 0} \frac{\mathbb{P}(t \leq T < t + \delta \mid T \geq t)}{\delta} \tag{11}$$

This results in the relationship: $S(t) = exp -\lambda(t)$, where $\Lambda(t) = \int_0^t \lambda(s)ds$ is the cumulative hazard function. The most common semi-parametric approach for estimating the hazard function is the Cox proportion hazards model (Cox, 1972), which assumes that the hazard function can be parameterized as an linear exponential function $\lambda(t \mid X) = \lambda_0(t) \exp\left(\beta^\top X\right)$ where the baseline hazard function $\lambda_0(t)$ describes how the risk of an event changes over time, $\beta$ are model parameters that describe how the hazard varies with features $X$. The baseline hazard $\lambda_0(t)$ is unspecified in the original model, making it difficult to estimate $\beta$, however, the Cox partial log-likelihood technique (Wong, 1986) can first estimate $\beta$ by maximizing:

$$\mathcal{L}_n(\beta) = \frac{1}{n} \sum_{i=1}^n \Delta_i \left[ \beta^\top X_i - \log\left( \sum_{j=1}^n Y_j\left(O_i\right) e^{\beta^\top X_j} \right) \right] \tag{12}$$

where $n$ is the set of patients, for the $i$-th subject in the study, $T_i$ and $C_i$ denote, respectively, the event time and the potential censoring time; and $X_i \in \mathbb{R}^p$ denotes the observed features. Thus, the observed data from a typical survival study contain independent observations $\mathcal{D} = \{X_i, O_i, \Delta_i\}_{i=1}^n$, where the observed time $O_i = \min\left(T_i, C_i\right)$ and the event indicator $\Delta_i = 1$, if the observed time is $T_i$, i.e. $T_i \leq C_i$, otherwise $\Delta_i = 0$, the subject is censored at $C_i$. Once the parameter $\beta$ has been estimated through the log partial likelihood, the cumulative baseline hazard function $\Lambda(t) = \int_0^t \lambda(s)ds$ can be estimated through the Breslow estimator (Breslow, 1972).

### B.2  C-INDEX

Due to the presence of censoring in survival data, traditional performance measures such as mean squared error cannot be used to evaluate the accuracy of predictions. Instead, the concordance-index

(C-index) (Harrell et al., 1982) is one of the most widely used performance measures for survival models. It assesses how good a model is by measuring the concordance between the rankings of the predicted event times and the true event times. Specifically, if the predicted event time of the $i$-th individual is $\hat{T}_i$, the C-index is defined by $C = \mathbb{P}\left(\hat{T}_i < \hat{T}_j \mid O_i < O_j, \Delta_i = 1\right)$. However, it is difficult to obtain the predicted event time in most survival models, so the following $C$-index proposed in Antolini et al. (2005) is often used in practice:

$$C = \mathbb{P}\left(\hat{S}\left(O_i \mid X_i\right) < \hat{S}\left(O_i \mid X_j\right) \mid O_i < O_j, \Delta_i = 1\right) \tag{13}$$

If $\{X_i, O_i, \Delta_i\}_{i=1}^n$ and $\hat{S}(t \mid X_i)$ denote observations and predicted conditional probabilities, respectively, the C-index in Eq. (13) can be estimated empirically by

$$\hat{C} = \frac{\sum_{i=1}^n \sum_{j=1}^n \Delta_i I\left(\hat{S}\left(O_i \mid X_i\right) < \hat{S}\left(O_i \mid X_j\right)\right)}{\sum_{i=1}^n \sum_{j=1}^n \Delta_i I\left(O_i < O_j\right)}.$$

The range of the C-index is $[0, 1]$, and larger values indicate better performance with a random guess leading to a C-index of $0.5$.

## B.3 WSI REPRESENTATION LEARNING

It has been shown that the WSI visual representations extracted by self-supervised learning methods on histopathological images are more accurate and transferable than the supervised baseline models on domain-irrelevant datasets such as ImageNet. In this work, a pre-trained Vision Transformer (ViT) model (Wang et al., 2021a) that is trained on a large histopathological image dataset has been utilized for tile feature extraction. The model is composed of two main neural networks that learn from each other, i.e. student and teacher networks. Parameters of the teacher model $\theta_t$ is updated using the student network with parameter $\theta_s$ using update rule represented in Eq. (14).

$$\theta_t \leftarrow \tau\theta_t + (1 - \tau)\theta_s \tag{14}$$

Two different views of a given input H&E image $x$, uniformly selected from training set $\mathcal{I}$, are generated using random augmentations, i.e. $u$, $v$. Then, student and teacher models generate two different visual representations according to $u$ and $v$ as $y_1 = f^{\theta_s}(u)$ and $\hat{y}_2 = f^{\theta_t}(v)$, respectively. Finally, the generated visual representations are transformed into latent space using linear projection as $p_1 = g^{\theta_s}\left(g^{\theta_s}(y_1)\right)$ and $\hat{z}_2 = g^{\theta_t}(\hat{y}_2)$ for student and teacher networks, respectively. Similarly, feeding $v$ and $u$ to student and teacher networks leads to $y_2 = f^{\theta_s}(v), \hat{y}_1 = f^{\theta_t}(u), p_2 = g^{\theta_s}\left(g^{\theta_s}(y_2)\right)$ and $\hat{z}_1 = g^{\theta_t}(\hat{y}_1)$. Finally, the symmetric objective function $L_{loss}$ is optimized through minimizing the $\ell_2 -$ norm distance between student and teacher as Eq. (15)

$$L_{loss} = \frac{1}{2}L(p_1, \hat{z}_2) + \frac{1}{2}L(p_2, \hat{z}_1) \tag{15}$$

where $L(p, z) = -\frac{p}{\|p\|_2} \cdot \frac{z}{\|z\|_2}$ and $\|\cdot\|_2$ represents $\ell_2 -$ norm.

## B.4 AGGREGATE PATCH LEVEL FEATURE INTO SLIDE LEVEL FEATURE

Survival outcome information is available at the patient-level instead of for individual slides, we use the attention based strategy in Porpoise (Chen et al., 2022b) which was originally designed in CLAM (Lu et al., 2021) to aggregate the patch level feature into slide level representation for our model training. We treat all WSIs corresponding to a patient case as a single WSI bag during training and evaluation. If a patient has $N$ WSIs with bag sizes $M_1, \cdots, M_N$ respectively, the WSI bag corresponding the patient is formed by concatenating all $N$ bags, and has dimensions $M \times 2048$, where $M = \sum_{i=1}^N M_i$.

There are three components for the model: 1) the projection layer $f_p$; 2) the attention module $f_{attn}$; 3) the prediction layer $f_{\text{pred}}$. After the VIT feature extraction, each patch-level embeddings of WSI bag, $\mathbf{H} \in \mathbb{R}^{M \times 2048}$, are first mapped into a more compact, dataset-specific 512-dimensional feature

space by the fully-connected layer $f_p$ with weights $\mathbf{W}_{\text{proj}} \in \mathbb{R}^{512 \times 2048}$ and bias $\mathbf{b}_{\text{bias}} \in \mathbb{R}^{512}$. Subsequently, the attention module $f_{\text{attn}}$ learns to score each patch for its perceived relevance to patient-level prognostic prediction. Patches receives high attention scores will contribute more to the patient-level feature representation than patches assigned low attention scores, all the patches in one patient's WSIs are aggregated based on attention-pooling (Ilse et al., 2018). Specifically, $f_{attn}$ has 3 fully-connected layers with weights $\mathbf{U}_a \in \mathbb{R}^{256 \times 512}$, $\mathbf{V}_a \in \mathbb{R}^{256 \times 512}$ and $\mathbf{W}_a \in \mathbb{R}^{1 \times 256}$. Given a patch embedding $\mathbf{h}_m \in \mathbb{R}^{512}$ (the $m^{\text{th}}$ row entry of $\mathbf{H}$ ), its attention score $a_m$ can be computed by:

$$a_m = \frac{\exp\left\{\mathbf{W}_a\left(\tanh\left(\mathbf{V}_a\mathbf{h}_m^\top\right) \odot \text{sigm}\left(\mathbf{U}_a\mathbf{h}_m^\top\right)\right)\right\}}{\sum_{m=1}^{M}\exp\left\{\mathbf{W}_a\left(\tanh\left(\mathbf{V}_a\mathbf{h}_m^\top\right) \odot \text{sigm}\left(\mathbf{U}_a\mathbf{h}_m^\top\right)\right)\right\}}$$

Then the patient-level representations $\mathbf{h}_{\text{patient}} \in \mathbb{R}^{512}$ are computed based on the attention-pooling operation from the patch-level feature representations by attention scores as weight coefficients, where $\mathbf{A} \in \mathbb{R}^M$ is the vector of attention scores:

$$\mathbf{h}_{\text{patient}} = \mathbf{Attn} - \mathbf{pool}(\mathbf{A}, \mathbf{H}) = \sum_{m=1}^{M} a_m \mathbf{h}_m$$

The last fully-connected layer is used to learn a representation $\mathbf{h}_{\text{WSI}} \in \mathbb{R}^{1 \times 32}$, which is then used as input to our multimodal fusion.

## B.5  SPARSE NETWORK FEATURE INTERPRETATION

We use the Integrated Gradients attribution algorithm to rank the features in all layers. Inspired by PNET (Elmarakeby et al., 2021), to reduce the bias introduced by over-annotation of certain nodes (nodes that are member of too many pathways), we adjusted the Integrated Gradients scores using a graph informed function $f$ that considers the connectivity of each node. The importance score of each node $i$, $C_i^l$ is divided by the node degree $d_i^l$ if the node degree is larger than the mean of node degrees plus $5\sigma$ where $\sigma$ is the standard deviation of node degrees.

$$d_i^l = fan - in_i^l + fan - out_i^l$$

adjusted $C_i^l = f(x) = \begin{cases} \frac{C_i^l}{d_i^l}, & d_i^l > \mu + 5\sigma \\ C_i^l, & \text{otherwise} \end{cases}$

## B.6  CO-ATTENTION BASED PATHWAY VISUALIZATION

After we get the ranking of top genes and pathways, we adopted the co-attention survival model (MCAT) (Chen et al., 2021) to show the spatial visualization of genomic features. We trained MACT on all our TCGA datasets, MACT learns how WSI patches attend to genes when predicting patient survival. We define each WSI patch representation and pathway genomic features as $H_{bag}$ and $G_{bag}$. The genomic features are the gene list values from the top pathways of each TCGA dataset. The model uses $G_{bag} \in \mathbb{R}^{N \times d_g}$ to guide the feature aggregation of $H_{bag} \in \mathbb{R}^{N \times d_p}$ into a clustered set of gene-guided visual concepts $\widehat{H}_{\text{bag}} \in \mathbb{R}^{N \times d_p}$ , $d_g$ and $d_p$ represents the dimension for the pathway (number of genes involved in the pathway) and patch. Through the following mapping:

$$\text{CoAttn}_{G \to H}(G, H) = \text{softmax}\left(\frac{QK^\top}{\sqrt{d_p}}\right)$$

$$= \text{softmax}\left(\frac{\mathbf{W}_q GH^\top \mathbf{W}_s^\top}{\sqrt{d_p}}\right)\mathbf{W}_v H \to A_{\text{coattn}}\mathbf{W}_v H \to \widehat{H}$$

where $\mathbf{W}_q, \mathbf{W}_s, \mathbf{W}_v \in \mathbb{R}^{d_p \times d_p}$ are trainable weight matrices multiplied to the queries $G_{\text{bag}}$ and key-value pair ( $H_{\text{bag}}$ , $H_{\text{bag}}$ ), and $A_{\text{coattn}} \in \mathbb{R}^{N \times M}$ is the co-attention matrix for computing the weighted average of $H_{\text{bag}}$ . Here, $M$ represents the number of patches in one slide and $N$ represents number of pathways (We trained top five pathways, so $N = 4$ in our study).

**Interpretation**: For a single genomic pathway embedding $\mathbf{g}_n \in G$, the co-attention module scores the pairwise similarity for how much $\mathbf{h}_m$ attends to $\mathbf{g}_n$ for all $\mathbf{h}_m \in H_{\text{bag}}$, written as a row vector $[a_{n1}, a_{n2}, \ldots, a_{nm}] \in A_{\text{coattn.}}$. These attention weights are then applied element-wise to $H_{\text{bag}}$, which constructs a new WSI-level feature embedding $\widehat{\mathbf{h}}_n \in \mathbb{R}^{n \times 1}$ that reflects the biological function of $\mathbf{g}_n$. For example, if $g_n$ is a genomic embedding that expresses the underlying biological pathways responsible for tumor formation, $A_{\text{coattn}}$ computed by the co-attention layer would saliently localize image patches containing tumor cells as high attention, which then aggregates $\widehat{\mathbf{h}}_n$ as a WSI-level representation primarily containing tumor cells. We describe the set of high attention image patches that attend to a single genomic embedding $\mathbf{g}_n$ as a "gene-guided visual concept", in which patches that are similar in feature space to $\mathbf{g}_n$ would share similar phenotypic information. For $N$ pathway embeddings in $G_{bag}$, the co-attention module captures up to $N$ different pathway-guided visual concepts, which we visualizes as attention heatmap in Fig. 3 b.

## C EXPERIMENTS

### C.1 NETWORK ARCHITECTURE

**Sparse network for gene**: The final gene expression embedding is $\mathbf{h}_g \in \mathbb{R}^{1 \times 50}$.

**Pathology network**: The slide level image feature representation is passed through an image embedding layer and encodes the embedding as $\mathbf{h}_p \in \mathbb{R}^{1 \times 50}$.

**CNV + MUT network**: Similarly as the pathology network, the patient level CNV + MUT feature representation is passed through an FC embedding layer and encodes the embedding as $\mathbf{h}_c \in \mathbb{R}^{1 \times 50}$.

### C.2 EXPERIMENTAL DETAILS

**PONET**. The latent dimensionality of the factorized matrices $k$ is a very important tuning parameter. We tune $k = [3, 5, 10, 20, 30, 50]$ based on the testing C-index value (Appendix Fig. 5) and the loss of training and testing plot (Appendix Fig. 6) for each dataset. We choose $k$ to maximize the C-index value and also it should have stable convergence in both training and testing loss. For example, we choose $k = 10$ in TCGA-KIRP for the optimized results. We can see that in Appendix Fig. 5 the testing loss is quite volatile when $k$ is less than 10. Similarly, we choose $k = [20, 10, 20, 20, 10]$ for TCGA-BLCA, TCGA-KIRC, TCGA-LUAD, TCGA-LUSC, TCGA-PAAD, respectively.

The learning rate and the regularization hyperparameter $\lambda$ for the Cox partial likelihood loss are also tunable parameters. The model is trained with Adam optimizer. For each training/testing pair, we first empirically preset the learning rate to 1.2e-4 as a starting point for a grid search during training, the optimal learning rate is determined through the 5-fold cross-validation on the training set, C-index was used for the performance metric. After that, the model is trained on all the training set and evaluated on the testing set. We use 2e-3 through the experiments for $\lambda$. The batch size is set to 16, and epoch is 100. During the training process, we carefully observe the training and testing loss for convergence (Figure 4 in Appendix C.2). The server used for experiments is NVIDIA GeForce RTX 2080Ti GPU.

**CoxPH**. We only include the age and gender for the survival prediction. Using CoxPHFitter from lifelines [3].

**DeepSurv** [4]. We concatenate preprocessed pathological images features, gene expression, and copy number + mutant data in a vector to train the DeepSurv model. L2 reg = 10.0, dropout = 0.4, hidden layers sizes = [25, 25], learning rate = 1e-05, learning rate decay = 0.001, momentum = 0.9.

**Pathomic Fusion** [5]. We use the pathomicSurv model which takes our preprocessed image feature, gene expression and copy number + mutation as model input. $k = 20$, Learning rate is 2e-3, weight decay is 4e-4. Batch size is 16 and epoch is 100. Drop out rate is 0.25.

**GPDBN** [6]. Learning rate is 2e-3, batch size is 16, weight decay is 1e-6, dropout rate is 0.3, epoch is

---

[3]https://github.com/CamDavidsonPilon/lifelines

[4]https://github.com/czifan/DeepSurv.pytorch

[5]https://github.com/mahmoodlab/PathomicFusion

[6]https://github.com/isfj/GPDBN

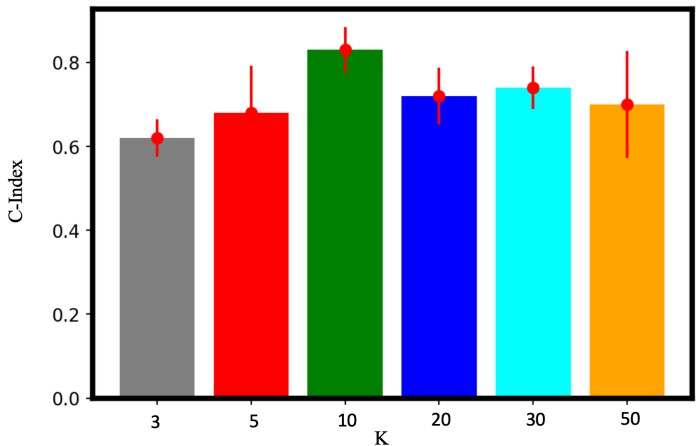

Figure 5: C-Index value under K = 3, 5, 10, 20, 30, 50 for TCGA-KIRP. The mean value and standard deviation for 5-fold cross-validation are plotted.

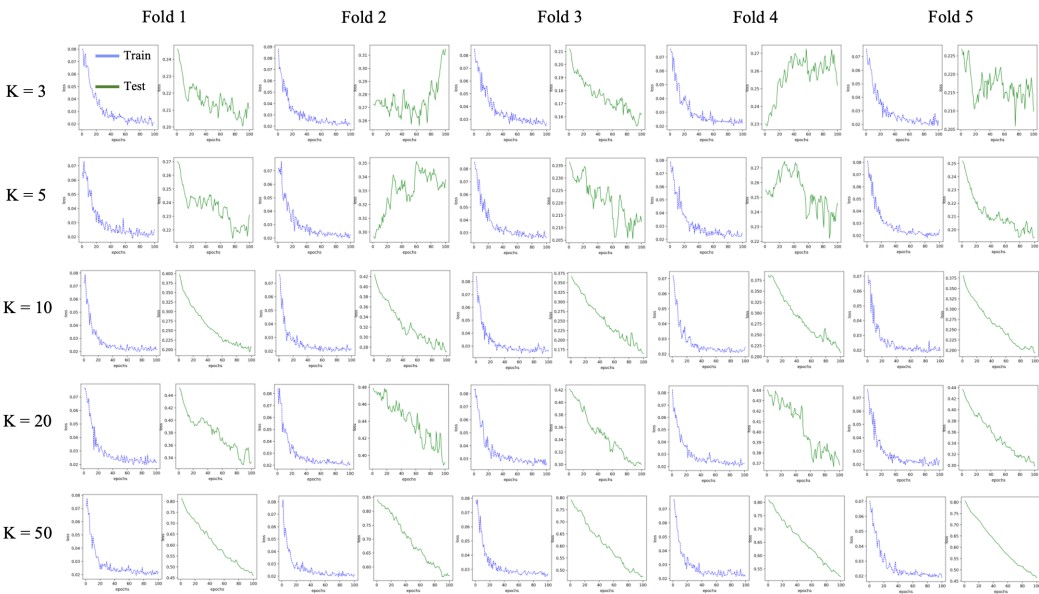

Figure 6: Train and test loss for TCGA-KIRP under K = 3, 5, 10, 20, 50 for 5-fold cross-validation.

100.

**HFBSurv** [7]. The learning rate is set to 1e-3, batch size is 16, $\lambda$ = 3e-3, weight decay is 1e-6, epoch is 100.

---

[7]https://github.com/Liruiqing-ustc/HFBSurv

## C.3   ADDITIONAL RESULTS

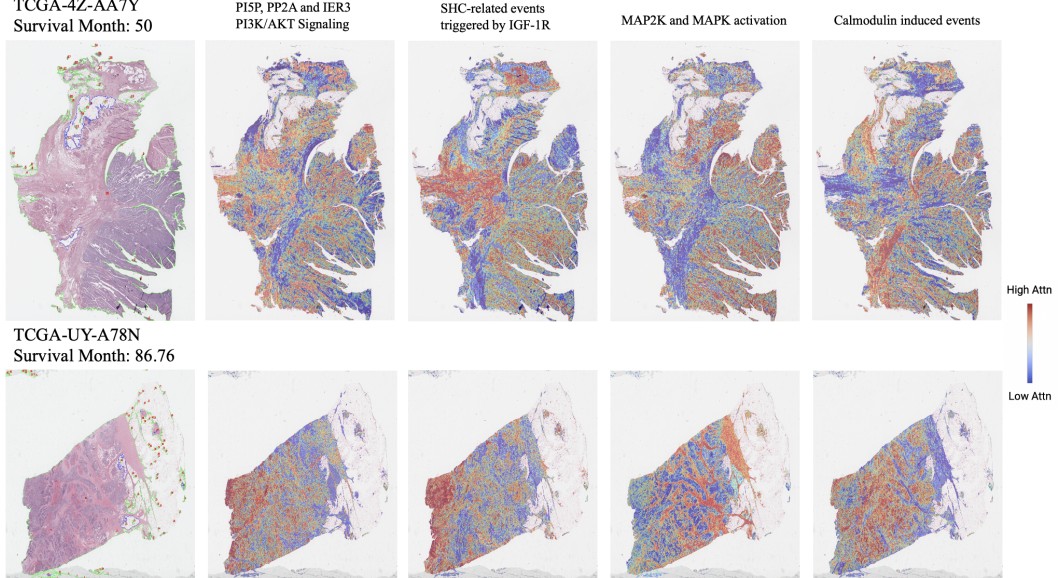

Figure 7: Inspecting and interpreting PONET on TCGA-BLCA. Sankey diagram visualization of inner layers of PONET shows the estimated relative importance of different nodes in each layer. Nodes in the first layer represent genes; the next layers represent pathways; and the final layer represents the model outcome. Different layers are linked by weights. Nodes with darker colours are more important, while transparent nodes represent the residual importance of undisplayed nodes in each layer.

Figure 8: Co-attention visualization of top 4 ranked pathways in two cases of TCGA-BLCA.

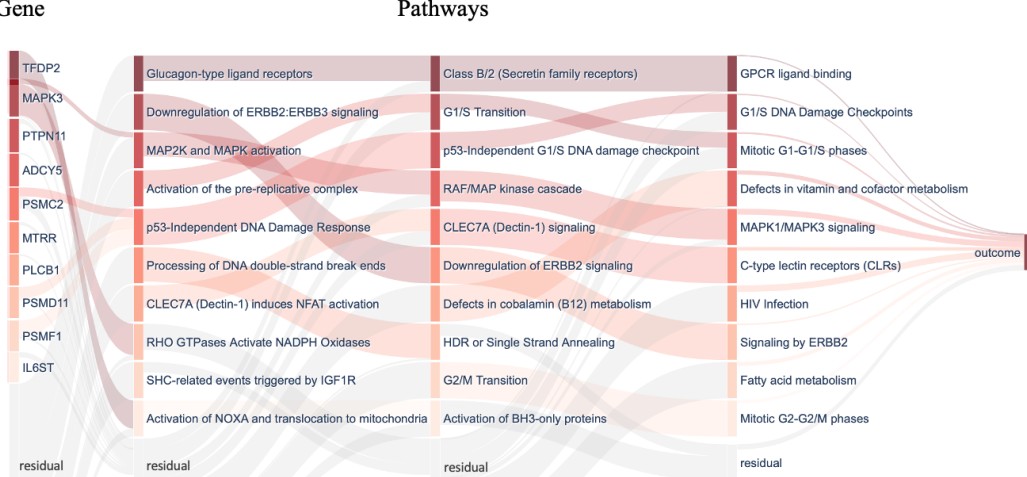

Figure 9: Inspecting and interpreting PONET on TCGA-KIRC. Sankey diagram visualization of inner layers of PONET shows the estimated relative importance of different nodes in each layer. Nodes in the first layer represent genes; the next layers represent pathways; and the final layer represents the model outcome. Different layers are linked by weights. Nodes with darker colours are more important, while transparent nodes represent the residual importance of undisplayed nodes in each layer.

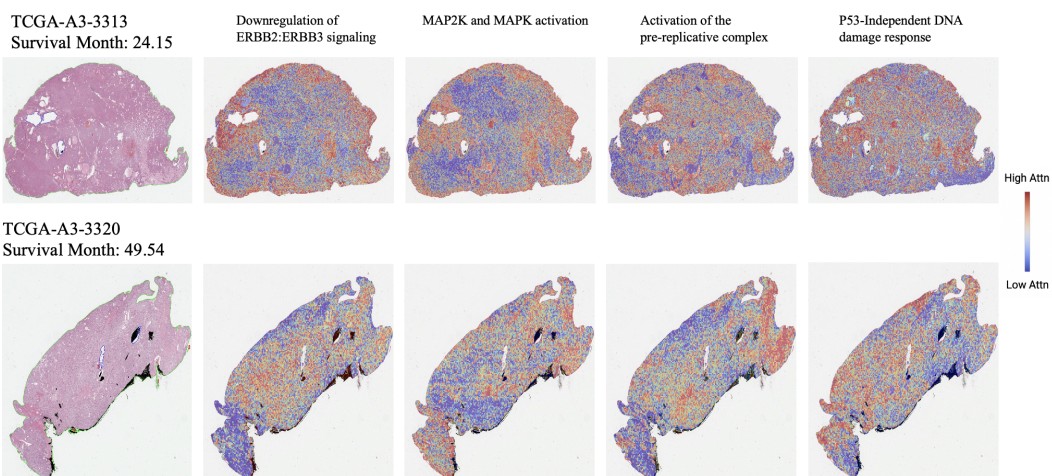

Figure 10: Co-attention visualization of top 4 ranked pathways in two cases of TCGA-KIRC.

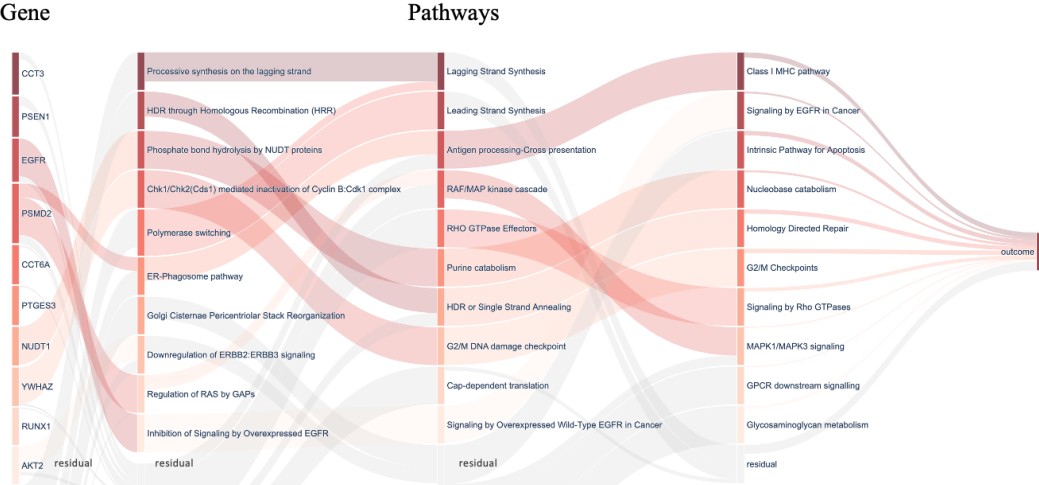

Figure 11: Inspecting and interpreting PONET on TCGA-LUAD. Sankey diagram visualization of inner layers of PONET shows the estimated relative importance of different nodes in each layer. Nodes in the first layer represent genes; the next layers represent pathways; and the final layer represents the model outcome. Different layers are linked by weights. Nodes with darker colours are more important, while transparent nodes represent the residual importance of undisplayed nodes in each layer.

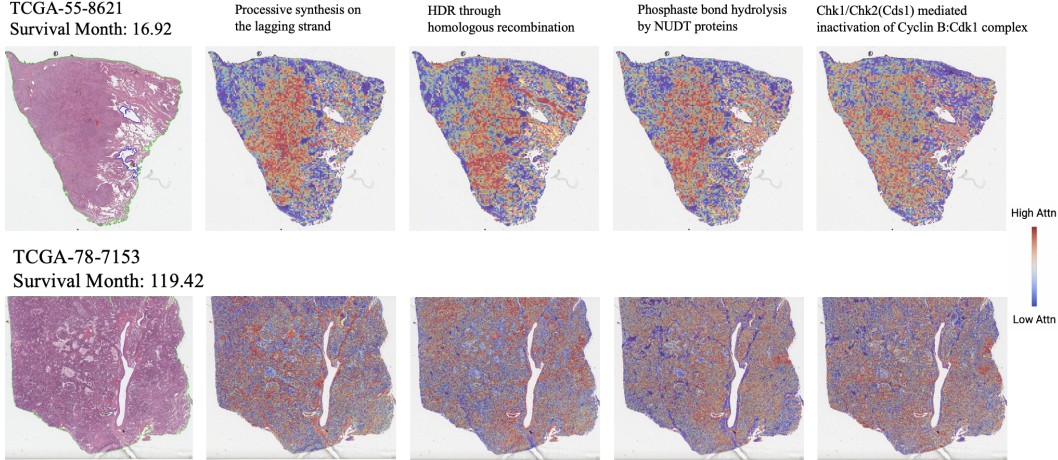

Figure 12: Co-attention visualization of top 4 ranked pathways in two cases of TCGA-LUAD.

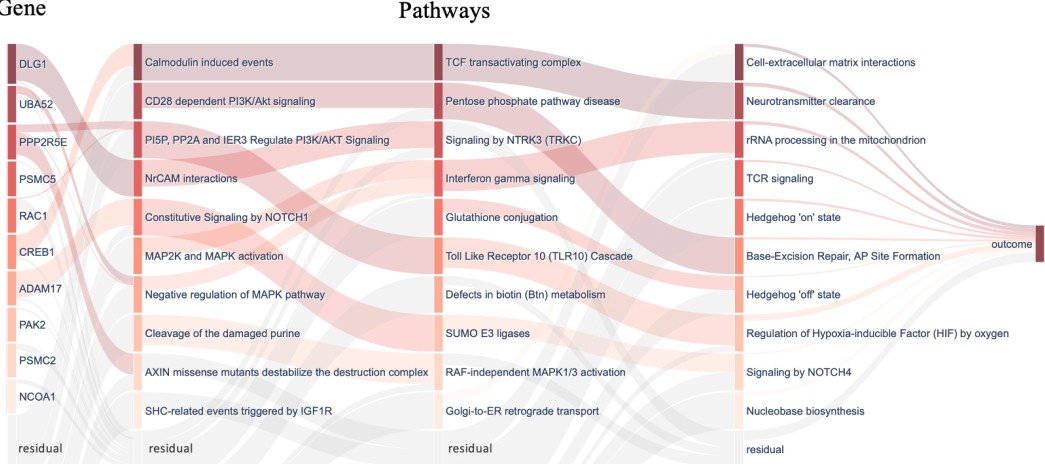

Figure 13: Inspecting and interpreting PONET on TCGA-LUSC. Sankey diagram visualization of inner layers of PONET shows the estimated relative importance of different nodes in each layer. Nodes in the first layer represent genes; the next layers represent pathways; and the final layer represents the model outcome. Different layers are linked by weights. Nodes with darker colours are more important, while transparent nodes represent the residual importance of undisplayed nodes in each layer.

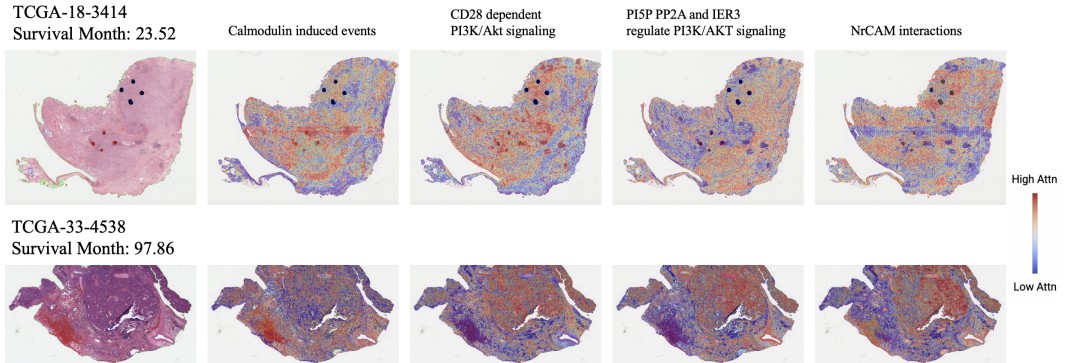

Figure 14: Co-attention visualization of top 4 ranked pathways in two cases of TCGA-LUSC.

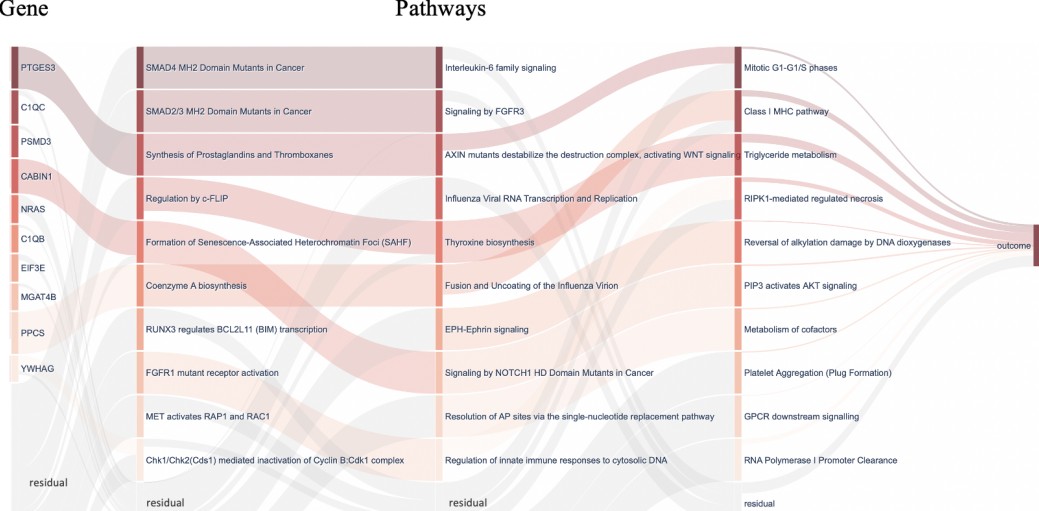

Figure 15: Inspecting and interpreting PONET on TCGA-PAAD. Sankey diagram visualization of inner layers of PONET shows the estimated relative importance of different nodes in each layer. Nodes in the first layer represent genes; the next layers represent pathways; and the final layer represents the model outcome. Different layers are linked by weights. Nodes with darker colours are more important, while transparent nodes represent the residual importance of undisplayed nodes in each layer.

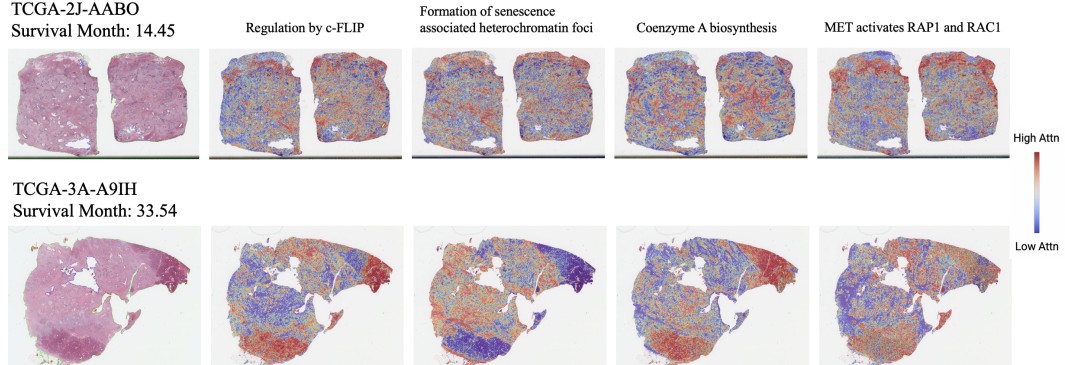

Figure 16: Co-attention visualization of top 4 ranked pathways in two cases of TCGA-PAAD.

