# OpenReview forum: "Deep Biological Pathway Informed Pathology-Genomic Multimodal Survival Prediction"
_ICLR.cc/2023/Conference — Submitted to ICLR 2023_

### Official Review · Reviewer_LXPk · 2022-10-17

**Confidence:** 3
**Correctness:** 3
**Technical Novelty And Significance:** 1
**Empirical Novelty And Significance:** 1
**Recommendation:** 5

**Clarity, Quality, Novelty And Reproducibility:**

$Clarity\ and\ Quality$:

Some phrases (names of modules) and statements need to modified. Overall, the paper writing is good and the contents are easy to understands. Few typos exist in the draft.

$Novelty$:

My major concern lies in the novelty. In my view, this paper neither proposes a novel pipeline for multi-modal survival prediction nor develops new modules for multi-modal fusion. Specifically, biological pathway-informed network has been used in similar problems (Han et al., 2018; Elmarakeby et al., 2021). The idea of hierarchical multi-modal fusion is borrowed from ARGF (Mai et al., 2020).

$Reproducibility$:

Lots of implementation details are provided. It is possible to reproduce the proposed method based on textual descriptions.




**Strength And Weaknesses:**

$Strengths$:

Overall, the descriptions of proposed PONET are clear and easy to understand. Lots of implementation details are included, making the proposed method easy to understand and improving the reproducibility. Pathway-based analysis is an interesting and promising direction for improving both predictive performance and biological interpretability. Besides, I appreciate the efforts made in visualizing the co-attention results.

$Weaknesses$:

1. The novelty is limited from perspectives of ideas and techniques. Multi-modal survival analysis is not a new idea (Mobadersany et al., 2018; Cheerla & Gevaert, 2019; Wang et al., 2020). Biological pathway-informed network has been used in similar problems (Han et al., 2018; Elmarakeby et al., 2021). The idea of attention-based multi-modal fusion methodology is borrowed from ARGF (Mai et al., 2020). The limited technical novelty is also reflected in Table 1. Comparing PONET with PONET-OH, we can draw a conclusion that the performance advantages of PONET mainly orginate from the incorporation of the pathway architecture. However, the pathway architecture has been used in similar problems (Han et al., 2018; Elmarakeby et al., 2021), and the authors did not mention the technical differences between their implementation and previous versions (Han et al., 2018; Elmarakeby et al., 2021), both of which weaken the technical contributions of PONET. Besides, I think the implementation details  of the sparse pathway network should be clarified in the main text instead of the appendix considering it is one of the core contributions.

2. Phrases like "Unimodal Fusion" and "Trimodal Fusion" are misleading and may exaggerate the contributions of this paper. "Unimodal Fusion" actually is equivalent to a feature embedding process, where the factorized multi-modal bilinear pooling (Yu et al., 2017; Li et al., 2022) is used to produce rich representations. Since it contains no interactions among different modalities, the use of "Fusion" is incorrect and misleading. A more appropriate name should be like "Unimodal Embedding". A similar problem also exists in "Trimodal Fusion", where the authors only apply the bimodal fusion strategy again. Personally, I recommend merging sec. 4.3 Bimodal Fusion and sec. 4.4 Trimodal Fusion into a single section as their operations are almost the same.

3. Experimental results are not sufficient. Since this paper heavily borrows the idea (i.e., hierarchical and attentional fusion) from ARGF (Mai et al., 2020) for multi-modal fusion, it is necessary to compare PONET with ARGF in the experiment section. However, I do not see any results of ARGF in the submission.  Besides, as shown in Table 2, bimodal and trimodal fusion are very useful. However, PONET only contains on bimodal fusion layer and one trimodal fusion layer. So, how about stacking more fusion layers? Also, the implementation details of single-fusion methods are not clarified, making the ablation results less convincing.

Minor points.

1. Notations of dimensions. For clarification, $x\in \mathbb{R}^m$ should be $x\in \mathbb{R}^{m\times 1}$. $y\in \mathbb{R}^n$ should be $y\in \mathbb{R}^{n\times 1}$.

2. In sec. 4.1, $m_1$, $m_2$ $\in (p, c, g)$ should be $m_1$, $m_2$ $\in$ {$p, c, g$}.

3. In sec. 4.3, "If two information vectors are close to each other, then little complementary information lies between them and their information has been well explored in the unimodal fusion." The use of "close" is inaccurate, how do you define the closeness between vectors? In addition, the unimodal fusion never explores the complementary information between two modalities' vectors regardless of their  similarity. Thus, the above statement should be rephrases.

4. All formulas need to be numbered.

**Summary Of The Paper:**

This paper presents PONET to fuse multi-modal data for making survival predictions. Specifically, a spare biological pathway-informed embedding network (Han et al., 2018; Elmarakeby et al., 2021) is used to deal with genomic data to reveal meaningful biological interpretations. To capture modality-specific information, the authors used bilinear pooling (Yu et al., 2017; Li et al., 2022), which provides richer representations than linear models. For multi-modal fusion, the idea from ARGF (Mai et al., 2020) is borrowed to assign attentions to identify the importance of cross-modal representations. On six cancer datasets from TCGA, PONET achieves performance improves over previous multi-modal survival prediction methods.

**Summary Of The Review:**

Based on the limited novelty of ideas and techniques, I'm afraid that the manuscript is not suitable for publish in its current version.

---

### Official Review · Reviewer_JgRd · 2022-10-23

**Confidence:** 5
**Correctness:** 3
**Technical Novelty And Significance:** 3
**Empirical Novelty And Significance:** 4
**Recommendation:** 3

**Clarity, Quality, Novelty And Reproducibility:**

Some part of originality pointed above is unclear.


**Strength And Weaknesses:**

Strength

- Strategy and model building sound.
- Experiments were conducted extensively.
- The proposed model showed better results compared to previously reported models.

Weaknesses

- If pathway informed is the key to the advantage of proposed model, how exactly the authors designed and inpremented such part of the model is unclear. How novel from Han et al. 2018 or Elmarakeby et al. 2021?
- Since the pathway informed strategy is tightly domain specific, it is quite difficult to generalize the authors finding to the other field, thus more specific venue might be suitable for meaningful discussion.


**Summary Of The Paper:**

The authors propose a novel biological pathway-informed hierarchical multimodal fusion model that intnegrates pathology image and genomic profile data for cancer prognosis.


**Summary Of The Review:**

The authors propose a novel biological pathway-informed hierarchical multimodal fusion model that intnegrates pathology image and genomic profile data for cancer prognosis. For several datasets from TCGA, the results indicated the advantage of the proposed model. Since the authors’ strategy is quite domain specific, it may be difficult to generalize the strategy.

---

> ### Author Response · Authors · 2022-12-08
> **Rebuttal feedback**
>
> Dear Reviewer JgRd,
>
> We also understand you are very busy and we appreciate your comments, and we tried our best to answer your questions in our rebuttal. In your initial comments, you argued that "Since the pathway informed strategy is tightly domain specific, it is quite difficult to generalize the authors finding to the other field, thus more specific venue might be suitable for meaningful discussion". We discussed the importance of our contribution to the multimodal survival analysis domain which is critical for biomarker discovery. And also the improved prediction performance will also make our method as a strong survival prediction method.  Please check the $\textbf{Response to all reviewers (x8M7, JgRd, and LXPk)}$.  Based on your suggestions, we also added our pathway design to our revised manuscript and conducted ablation studies to compare our pathway architecture with the two related literature. We hope you can re-evaluate the strengths of our work and raise the score if we addressed your concerns.
>
> Thank you very much!
>
> Best regards,

---

### Official Review · Reviewer_x8M7 · 2022-10-25

**Confidence:** 3
**Correctness:** 3
**Technical Novelty And Significance:** 2
**Empirical Novelty And Significance:** 2
**Recommendation:** 5

**Clarity, Quality, Novelty And Reproducibility:**

Clarity & Quality & Novelty
Please see “Strengths and Weaknesses”. I think the manuscript needs a carful revision.

Reproducibility
I think the manuscript has no issue with this item.


**Strength And Weaknesses:**

Strengths
1. PONET shows the supervisor prediction performance over the base-line methods, including the Kronecker product-based fusion methods and the MLB-based method. In addition, the proposed methods, which are based on the MLB, have lower complexities compared to the Kronecker product-based fusion methods
2. The methods provide biological interpretations which would be useful to identify new biomarkers markers from data.

Weakness
1. I did not see significant improvement in methodology when comparing the proposed methods to HFBsurv [Li et al., 2022]. It seems that all the main components (the way to fuse multi modalities, attention mechanism, and even all the main equations) are directly borrowed from the previous work.
2. It is a little bit hard to understand the meanings of several points. I think that the paper needs to be revised.
For example,
2-a) about the equation in the top of page 3, regarding the attention, requires the form of the attention. I don’t think this is the standard attention module which is based on queries, keys and values.
2-b) In the section 5.2 (comparison with Baselines), “All versions of PONET outperform Pathomic Fusion by ~“. The authors listed two main reasons, but it seems that they need more explanations. For the reason 1, it sounds reasonable that the proposed methods, based on the MLB (low rank), are less computational completed than Pathomic. However, it is not clear why the MLB-based-methods are necessarily outperforms the Kronecker-product-based methods. For the reason 2, are the authors trying to compare both feature extraction methods, VIT and GCN?
3. In the section 5.3, could the authors give more explanations about why discretization of Survival time can lead better performance for certain cancer types.
4. Regarding Figure 3-b. It is not clear how co-attention visualization was done. I thought that it could be done by postprocessing the outputs of the trained PONET. However, the appendix just includes a reference method. Could the author elaborate this process in the manuscript?
5. Figure 1 and the experiments. What kind of the pathway database is used to model the PONET pathway structure? I can see how the layer H1 (genes) and H2 (pathways) are structured (the pathway database indicates the member genes of each pathway), but don’t understand how to build the layers H3 and H4. These kinds of information should be included in the text.
6. I think the manuscript needs a careful proofreading. Some minor types grammatic errors. For example, in Page 3, “Unlike our proposed PONET can” might be “which“ was missing here?


**Summary Of The Paper:**

Deep Biological Pathway Informed Pathology-Genomic Multimodal Survival Prediction

The manuscript proposes a multi-model data integration method for survival prediction. The method, called PONET, integrates gene expressions, pathology image features and copy number and mutation data using the multi-model low-rank bilinear pooling (MLB, implemented by SumPooling) which is a variant of the bilinear models.  Specifically, PONET extends the previous work HFBSurv [Li et al., 2022] to include the trimodal fusion (combining two bimodal fusion outputs) and the pathway structures for hidden layers. The experimental results show that the proposed methods work better for survival prediction in several cancer types in TCGA cohort. In addition, the methods provide biological interpretations (e.g., gene-pathway associations or image-pathway associations using co-attention visualization)


**Summary Of The Review:**

I think that the manuscript still needs improvement to be accepted by the conference (mainly due to its limited novelty and presentation). However, I like the fact that the methods provide biological interpretations, specifically the information presented in Figure 3.

---

> ### Author Response · Authors · 2022-12-08
> **Rebuttal feedback**
>
> Dear Reviewer x8M7,
>
> We understand that you are very busy, and we really appreciate your constructive comments to help us revise our manuscript significantly. As you highlighted in your initial comments that "The methods provide biological interpretations which would be useful to identify new biomarkers markers from data". PONET is a novel framework to model the multimodal data for survival prediction and we do believe the potential practical use of PONET is huge in clinical studies. We tried our best to address your concerns and highlighted our novelties in $\textbf{Response to all reviewers (x8M7, JgRd, and LXPk)}$. We would like to know whether you are satisfied with those responses. If so, we want to sincerely check if you could re-evaluate our contributions and raise the score. Also if you have other concerns we would also like to discuss them. Thank you so much!
>
> Best regards,

---

### Decision · Program_Chairs · 2023-01-20

**Decision:**

Reject

**Justification For Why Not Higher Score:**

The authors are commendable for improving their work during discussion, nonetheless, given the reviewers' major concerns remain, it would be ideal for the author to consider enhancing this work further, and potentially this work can be a stronger candidate where the audience are more domain-specific to the topics being discussed.

**Justification For Why Not Lower Score:**

N/A

**Metareview: Summary, Strengths And Weaknesses:**

In this work, the authors present a method to integrate multi-model data for survival prediction. They leveraged sparse pathway-informed network that leverage both pathology images and genomic data to improve the performance and help to identify master genes/pathways for the prediction. The authors demonstrated their methods using TCGA cancer patient data. Overall this work is well-presented, clear, provide a very good framework for integrative ML-based survival prediction that provide meaningful interpretations. The major concerns raised were the ideas used were overall reasonably straightforward, and the applications are more limited to specific domain.